# Telomere Maintenance in Pediatric Cancer

**DOI:** 10.3390/ijms20235836

**Published:** 2019-11-20

**Authors:** Sandra Ackermann, Matthias Fischer

**Affiliations:** 1Department of Experimental Pediatric Oncology, University Children’s Hospital of Cologne, Faculty of Medicine and University Hospital of Cologne, Kerpener Straße 62, 50937 Cologne, Germany; 2Center for Molecular Medicine Cologne (CMMC), University of Cologne, Robert-Koch-Straße 21, 50931 Cologne, Germany

**Keywords:** telomere maintenance, pediatric cancer, neuroblastoma

## Abstract

Telomere length has been proposed as a biomarker of biological age and a risk factor for age-related diseases and cancer. Substantial progress has been made in recent decades in understanding the complex molecular relationships in this research field. However, the majority of telomere studies have been conducted in adults. The data on telomere dynamics in pediatric cancers is limited, and interpretation can be challenging, especially in cases where results are contrasting to those in adult entities. This review describes recent advances in the molecular characterization of structure and function of telomeres, regulation of telomerase activity in cancer pathogenesis in general, and highlights the key advances that have expanded our views on telomere biology in pediatric cancer, with special emphasis on the central role of telomere maintenance in neuroblastoma. Furthermore, open questions in the field of telomere maintenance research are discussed in the context of recently published literature.

## 1. Introduction

Precise control of genomic stability is essential for cellular integrity and tissue homeostasis in multicellular organisms. Stabilization of chromosomal ends by telomeric DNA is evolutionary conserved and was first described in the late 1930s in *Drosophila* by Herman Muller and in maize by Barbara McClintock [1,2]. The fundamentals of molecular telomere function were published by the 2009 Nobel Prize laureates Elizabeth Blackburn, Carol Greider and Jack Szostak, who demonstrated “how chromosomes are protected by telomeres and the enzyme telomerase” [3]. Today, telomere biology has been implicated in major human biological processes and conditions, such as reproduction, aging, cancer, cardiovascular disorders, various heritable syndromes, other human pathologies and even mental health [4,5,6], and many excellent comprehensive reviews on these topics are available, summarizing current knowledge and outlining promising future trends [4,5,7,8]. Still, deciphering the complexity of telomere biology is steadily increasing our knowledge of how telomeres function to maintain genome integrity and how their dysfunction gives rise to human disease. In this review, we focus on the latest progress in understanding the molecular nature of telomere biology in cancer with particular emphasis on pediatric oncology, and highlight promising findings that may provide the basis for development of molecular diagnosis and novel approaches to rational treatment strategies.

## 2. The Role of Telomeres in Genome Stability

Telomeres are non-coding regions of repetitive DNA sequences (TTAGGG in vertebrates) at the end of each chromosome arm of eukaryotic linear chromosomes [9]. In combination with a protein complex called shelterin, these sequences form a cap which provides at least two protective functions: (1) It allows cells to distinguish chromosome ends from double-strand breaks; (2) It protects chromosomes from end-to-end fusion, recombination and degradation [10,11]. Over the past three decades, the lack of precise structural information on telomeric shelterin complexes and of atomic details of TERT, the catalytic subunit of telomerase, have been major bottlenecks in fully understanding molecular properties of telomere maintenance. Recent advances in crystallography instrumentation and image processing have dramatically improved the resolution and quality of crystallographic models, providing detailed insights into the structure of the catalytic core alone and in interaction with its nucleic acid substrates [12,13].

## 3. Telomeres in Aging and Disease

In human somatic cells, the average telomere length typically ranges from 10 to 15 kb and shortens at a rate of 50–200 bp with each cell division (“end-replication problem”) [14]. In the absence of telomere maintenance mechanisms, average cells divide between 50 and 70 times before the loss of chromosome capping function at telomeres leads to replicative senescence or programmed cell death (“Hayflick phenomenon”) [15]. Hence, telomere length has been widely considered as the “mitotic clock” and used as a marker of biological ageing [16,17,18]. Meanwhile, this historical view has expanded and telomere length is now considered to be a far more complex marker that takes into account cumulative effects [19]. While leukocyte telomere length has been shown to be significantly associated with all-cause mortality, no study has provided evidence that telomere length can reliably predict function decline in old people, or may actually predict age, morbidity or mortality in individuals [20,21]. Telomere shortening is a weak biomarker with poor predictive accuracy for personalized medicine and does not provide an independent clinical insight in individual age, age-related disease susceptibility or frailty, but is a rather complex accompanying phenomenon mirroring a kaleidoscope of environmental and genetic factors in addition to chronological age [20,22,23,24,25].

Both extremely short and long telomeres have been observed not only in cancer cells, but in various diseases that vary in severity and span the entire age spectrum from infancy to adulthood [26,27]. The short telomere phenotype has been linked to a broad spectrum of clinical conditions such as increased risk of cardiovascular disease [28,29], heart failure [28], cancer [30,31], diabetes [32,33], T cell immunodeficiency [34], schizophrenia [35], depression [36], decline in cognitive function [37,38] and early mortality [39]. Even though all these conditions associated with short telomeres may intuitively suggest that long telomeres confer an advantage for health and lifespan, there is strong evidence that this view may be too trivial. Longer telomeres appear to increase the risk of numerous cancer types [7], especially melanoma [40], adult glioma [41], and chronic lymphocytic leukemia [42,43]. Similar analyses in childhood cancers have observed associations between longer telomeres and the risk of neuroblastoma [44] and osteosarcoma [45].

The first causal connection between telomeres and human disease was found when germline mutations in the gene *DKC1* (dyskerin) were detected in dyskeratosis congenital, a rare inherited X-linked premature aging syndrome [46,47]. Dyskerin is a core component of the telomerase complex, and missense mutations in this gene result in telomerase enzyme deficiency, bone marrow failure and progressive telomere shortening [47]. Nowadays, dysregulation of telomere maintenance is known to be implicated in a broad spectrum of medical conditions way beyond aging syndromes. Apart from bone marrow failure and lung disease, the most common telomere spectrum disorders include conditions like squamous cell cancer of the skin, oral leukoplakia, pulmonary fibrosis and gastrointestinal diseases such as esophageal stricture, enteropathy, and enterocolitis [48,49,50,51,52,53,54]. Clinical observations in patients with telomere syndromes also shed light on the role of telomeres in cancer. Just like other DNA repair disorders, inherited telomere disorders are cancer prone, but the overall incidence appears to be relatively low (reviewed in [55]).

## 4. Telomere Maintenance in Cancer

Cancer is a leading cause of death worldwide with advancing age being the most significant risk factor. In 2018, there were 18.1 million cancer cases worldwide and 9.6 million deaths. Cancer cases increased by 35% between 2006 and 2018 [56]. Today, it is generally established that cancer cells are typically characterized by chaotic karyotypes, genome instability and immortality as a result of acquired mechanisms to circumvent normal replicative barriers [57,58,59,60]. Therefore, telomere shortening has two opposing effects during cancer development: Once telomere degradation has reached a critical level, the cell is committed to proliferation-limiting pathways that ultimately result in senescence or apoptosis. On the other hand, if this restriction fails, loss of telomere protection can induce genomic instability and numerous genomic changes, such as chromothripsis, kataegis and tetraploidization that ultimately result in malignant transformation [8,61].

Maintaining telomere length above a critical threshold is essential for immortalization of malignant cells and a critical factor in tumorigenesis in both adult and pediatric cancers [62,63]. Telomere maintenance in cancer can be achieved by two major mechanisms: Induction of telomerase, or a mechanism termed alternative lengthening of telomeres (ALT). In cells with active telomerase, the enzyme preferentially targets short telomeres, but telomere length is continually being degraded and elongated, thereby retaining telomere length homeostasis and functionality [64]. There seems to be no selective advantage for cancer cells having more telomerase activity than is needed to protect against DNA-damage signaling or end-fusion [31,65,66,67]. In line with this, systematic analysis of telomere length across 31 cancer types revealed that 70% of all samples exhibited shorter telomeres compared to normal samples [63]. Comparable results were obtained in a recent study analyzing samples from 653 patients with 23 cancer types from the Pediatric Cancer Genome Project, with shorter telomeres compared to normal samples in the majority of cases [68]. In cancer cells lacking telomerase, telomere maintenance is achieved by ALT, which is a telomerase-independent, recombination-based mechanism [31,63]. It is important to note, however, that telomere maintenance in itself is not sufficient for malignant transformation, which requires additional events leading to activation of oncogenes and/or inactivation of tumor suppressor genes, such as *TP53* or *RB1* [69,70]. Non-malignant cells expressing ectopically introduced *TERT* exhibit normal cell-cycle activities, maintain contact inhibition and anchorage-dependent growth requirements, and maintain a normal karyotype [71].

## 5. Telomerase Activation

Telomerase activity enables cells to overcome replicative crisis or senescence and thus provides the cell with the capacity to divide infinitely [72,73,74,75]. The basic molecular mechanisms of telomerase activity are highly conserved throughout evolution and our understanding of the holoenzyme complex is steadily improving since its discovery by Blackburn and Greider in 1985. In yeast, animals, and plants, telomerase consists of the telomerase reverse transcriptase (TERT) protein providing the catalytic activity, and the telomerase RNA template subunit (TERC), which serves as a template for the synthesis of telomere sequences [76]. Telomerase activity is well correlated with *TERT* expression, and ectopic expression of *TERT* expression in telomerase negative cells is sufficient to confer telomerase activity, suggesting that TERT is the rate-limiting component of telomerase activity in most cells with few exceptions, such as the developing embryo during gestation [77,78,79,80,81,82,83]. The significance of telomerase activation in tumorigenesis has been substantiated by several *TERT*-transgenic mouse models, in which constitutive telomerase expression led to an increased incidence of spontaneous tumors (reviewed in [84]). Apart from the well-established enzymatic function of telomerase, emerging evidence points towards non-telomeric roles of both TERT and TERC in tumorigenesis via regulation of tRNAs and tRNA derivatives, as well as activation or stabilization of multiple cancer relevant processes, such as Wnt-, ATM/ATR-, myc- and NF-kappa-B signaling pathways [85,86,87,88].

With the exception of highly proliferative cells within self-renewing tissues, *TERT* expression is essentially absent in the majority of cells in normal tissues [89,90]. In addition, telomerase activation in somatic cells is transient, which is in sharp contrast to constitutive telomerase activation in most cancers (reviewed in [91]). Systematic analysis of 6,835 tumor samples from 31 different human cancers revealed that *TERT* expression was present in 73% of all samples, which was associated with genomic alterations of *TERT*, such as point mutations, rearrangements, DNA amplifications and transcript fusions [63]. Similar observations have been made in studies focusing on pediatric tumors, although profound differences in *TERT* expression levels were found in different cell and tissue types [68,92,93].

During the last two decades, substantial advances have been made in understanding the molecular mechanisms causing *TERT* upregulation in cancer [63,94,95,96]. Since the first report of highly recurrent mutations in the *TERT* promoter region in melanoma [97,98], their presence has been reported in numerous cancers [94]. Hotspot mutations at locations -124 (C > T) and -146 (C > T) were reported to be the most common point mutations and often associated with poor prognosis, with -124 C > T being dominant over -146 C > T [99]. These genetic alterations create DNA binding sites that specifically recruit the multimeric E26 transformation-specific factor GABP, leading to increased *TERT* expression [100,101,102]. A survey of 1,230 samples of 60 different cancer types revealed tumors with low (<15%) and high (≥15%) frequencies of *TERT* promoter mutations. High mutation frequencies were found in nine cancer types, including melanoma, liposarcoma, hepatocellular carcinoma, urothelial carcinoma, squamous cell carcinoma of the tongue, medulloblastoma, and adult glioma subtypes (including 83% of primary glioblastoma, the most common brain tumor type). In contrast, the frequency of *TERT* promoter mutations was considerably lower (11%) in primary glioblastomas of pediatric patients [103]. In general, *TERT* promoter mutations appear to mainly represent characteristic alterations of adult cancers, and their occurrence is strongly correlated with age [104]. In pediatric cancers and tumors of young adults, these alterations occur at a much lower overall-frequency of about 2.5%, with the exception of sonic-hedgehog-activated (SHH) medulloblastoma, especially those tumors diagnosed in adolescents and young adults [75,96,104,105,106,107,108,109]. Data on age distribution of *TERT* promoter mutations in pediatric cancers is rare, but a few studies suggest a lower frequency in infants as compared to older children [104,110]. Likewise, polymorphisms in telomere maintenance-associated genes have been suggested to predispose to cancer and have been detected frequently in adult but rarely in pediatric tumors [111].

*TERT* activation may also be conferred by genomic rearrangements of the *TERT* locus, which enhance local DNA accessibility by causing massive changes of the chromatin landscape. These alterations have been reported in both adult and pediatric solid tumors [112,113,114]. The majority of such rearrangements result in translocation of strong enhancer regions from other genes to the *TERT* coding sequence, thereby inducing *TERT* expression much more efficiently than *TERT* promoter mutations [63,113,115]. In addition, genomic amplification of the *TERT* locus is a common mechanism of telomerase activation in some cancer types, particularly hepatocellular carcinoma (15%), lung adenocarcinoma (18%) and squamous cell carcinoma (25%), colorectal carcinoma (48%), and cervical intraepithelial neoplasms 2 and 3 (60% and 88%, respectively) (reviewed in [116]). Overall, however, only 3% of all *TERT* expressing tumor samples present with *TERT* amplification [63,93].

Still, the mechanisms of telomerase activation have remained unclear in a substantial number of human cancers that lack the aforementioned alterations (e.g., certain pediatric tumors of the nervous system, and prostate, pancreatic and gastric cancer, among others) [93,103,104,116,117]. Thus, there is increasing interest in the role of epigenetic and transcriptional regulation of *TERT* in cancer (reviewed in [118]). In fact, it has been demonstrated that more than 50 trans-acting proto-oncogenes and other factors are able to directly interact with the genomic region between 3.5 kb upstream and 150 bp downstream of the translation start site of *TERT*, thereby contributing to the complex regulatory network involved in controlling the *TERT* expression [119]. For example, transcription factors of the MYC family (c-MYC and MYCN), which are associated with aggressive pediatric cancers such as medulloblastoma and neuroblastoma, bind to and activate the *TERT* promoter, resulting in high *TERT* mRNA levels of *MYC-* and *MYCN*-amplified tumors [113,120,121,122].

Transcription factor binding is important but may not be sufficient to govern *TERT* expression [123]. Alterations of the chromatin environment, such as histone modification patterns, are also contributing to the activity of the *TERT* locus [93,103,104,116,117,124]. Promoter methylation, for example, is often associated with gene silencing by preventing transcriptional modulators from binding to the DNA [125,126]. However, *TERT* was actually one of the first genes in which methylation of its promoter sequence had been found to have an opposite effect and was positively correlated with gene expression [127]. This hypermethylation pattern occurs in a multitude of both pediatric and adult cancer types, including neuroblastoma, medulloblastoma, prostate, gastric, esophageal, cervical, and colorectal cancer, but the precise mechanisms by which *TERT* promoter methylation regulates *TERT* expression are still under investigation and suggested to be tissue/cell-type-specific [93,128,129,130,131,132,133,134]. One hypothesis (and non-mutually exclusive mechanism) is that promoter hypermethylation can open the promoter proximal chromatin conformation, due to the high GC content of the *TERT* promoter and its propensity to adopt G-quadruplex conformations, and thus result in increased transcription of the locus [93,135,136].

Furthermore, additional modulating factors such as micro RNAs, long non-coding RNAs, transposable elements, certain common genetic variants and viruses have been described to regulate telomerase biogenesis, subcellular localization and function, but covering these aspects would go beyond the scope of this review [25,137,138,139].

Taken together, a multitude of molecular mechanisms have been described that upregulate telomerase in cancer cells. Reliable and sensitive detection of biomarkers indicating telomerase-dependent telomere maintenance, however, is not trivial. Several studies have shown that, besides measuring telomerase activity directly, *TERT* expression may be a useful surrogate marker for evaluation of telomerase-dependent telomere maintenance, although it has to be taken into account that measurement of a continuous variable, such as gene expression, always has a certain risk of calling false-positives and false-negatives (reviewed in [140]).

## 6. Alternative Lengthening of Telomeres (ALT)

While most cancers rely on active telomerase, a lower but significant proportion utilizes the recombination-dependent alternative lengthening of telomeres (ALT) pathway. ALT was originally discovered in immortalized cell lines and subsequently shown to occur in human tumors [141,142]. While the mechanism of ALT has not been completely elucidated yet, several hallmarks of this telomere maintenance pathway have been defined. Apart from a lack of telomerase activity, these characteristics include longer telomeres on average (∼20 kb) than in telomerase-positive cells, a more heterogeneous telomere length distribution within individual cells and across tumor cell populations, accumulation of extrachromosomal dsDNA and ssDNA structures termed T-circles [143] and C-circles [144] or extrachromosomal telomeric repeats (ECTRs), the presence of variant telomeric repeat sequences, elevated rates of telomeric-sister chromatid exchange (T-SCE), mutations in specific genes such as *ATRX*, *DAXX* and *SMARCAL1*, high levels of the non-coding telomeric RNA transcript TERRA, and formation of ALT-associated promyelocytic leukemia (PML) nuclear bodies (APBs) [145,146,147,148,149].

Although recombination has been thought to play a major role at telomeres for many years, characterization of the molecular assembly of essential factors and execution of telomeric sequence replication happening at sites of homology-directed DNA repair is still ongoing [150,151,152]. It has been hypothesized that a specialized replisome underlies telomere synthesis in ALT, and that this pathway can be mediated by two distinct mechanisms [153,154]. One is a RAD51-dependent/RAD52-independent recombination-mediated process leading to preferential elongation of lagging-strand overhangs, which is mechanistically similar to the yeast type I ALT. The other one is similar to the yeast type II ALT, a RAD51-independent/RAD52-dependent mechanism mediated by POLD3-dependent break-induced replication. However, most of the molecular details about these two pathways have remained unexplored until very recently [155,156].

Common methods for detecting an ALT phenotype in tumor samples are assays for APBs [145], C-circles [144], or telomere length [142,157]. Recent analysis of large cohorts has begun to reveal the prevalence of ALT in human cancers, highlighting the enrichment of ALT in tumors arising from neuroendocrine systems, mesenchymal and neuroectodermal cells including bone, soft tissues, peripheral and central nervous systems (reviewed in [158]). The prevalence of this telomerase-independent pathway ranges from 25% to 60% in sarcomas and 5% to 15% in carcinomas [157,159]. In a recent study analyzing samples from 653 patients with 23 cancer types from the Pediatric Cancer Genome Project, telomere lengthening indicative of ALT was observed in 28.7% of solid tumors, 10.5% of brain tumors, and 4.3% of hematological cancers [68]. It has to be mentioned, though, that estimations and comparisons from different methods and analyses are still challenging due to differences in laboratory techniques, data configuration and normalization [158,160]. Although low levels of ALT specific C-circles have been found in certain human healthy tissues, such as placenta in early gestation, in general, ALT is thought to be repressed in normal human somatic cells and several studies have identified factors in this regulatory system [157,161,162,163]. It has remained unclear to date, whether activation of telomerase and ALT within the same cell or populations can co-exist, and whether switching between the two mechanisms may occur [158,160].

Taken together, while several questions related to molecular mechanisms driving telomere maintenance have remained unresolved, it is clear that cancer cells essentially depend on telomere maintenance for immortalization and malignant behavior, and an absence or low frequency of cellular immortalization is observed in most benign neoplasms and normal tissues [116,159,164].

## 7. Telomere Maintenance in Pediatric Oncology

Childhood cancers are rare and represent less than 1% of all cancers. The overall incidence rates of childhood cancer vary between 50 and 300 per million children across the world [165,166]. Unlike most cancers in adults, many childhood tumors are not strongly linked to lifestyle or environmental risk factors [167,168]. Not surprisingly, the mutational landscapes and the prevalence of telomere maintenance mechanisms in adult and pediatric tumors differ substantially, thus emphasizing the need to consider adult and pediatric tumors separately [109,169]. For example, in hepatoblastoma, the most common childhood liver cancer, as well as in pediatric gliomas, acute myeloid leukemia and thyroid cancers, *TERT* gene or promoter mutations have not been observed or far less frequently than in corresponding adult malignancies [104]. On the other hand, higher frequencies of ALT-positive tumors have been reported [104,159,169,170,171,172,173,174,175]. It has to be noticed, however, that despite growing interest in telomere maintenance mechanisms in pediatric cancer, many studies have examined only few samples of each individual tumor type (Table 1), so caution is warranted in drawing final conclusions. In Table 1, results on the prevalence of telomerase and ALT activation obtained from analyses of initial pediatric tumor samples are summarized [60,62,174,176,177,178,179,180,181,182,183,184,185,186].

## 8. Neuroblastoma

In terms of telomere biology, neuroblastoma has been more extensively studied than any other childhood cancer [106,113,115,174,175,187,188,189,190,191]. This malignant embryonic tumor arises from precursor cells of the sympathetic nervous system and is the most common solid extracranial malignancy of childhood [192]. The biological and clinical phenotypes of neuroblastoma are remarkably heterogeneous, ranging from spontaneous regression in low-risk tumors to treatment-resistant fatal progression despite aggressive therapies in high-risk disease. Amplification of the proto-oncogene *MYCN,* a known transcriptional activator of *TERT*, occurs in roughly 20% of primary neuroblastomas and is a well-established marker for defining high-risk disease. More recently, several other genomic alterations involved in telomere maintenance have been identified by massively parallel sequencing studies, and are considered to be indicators of poor prognosis [115,193,194]. A key finding was the discovery of recurrent genomic rearrangements of the *TERT* locus, which were consistently associated with massive transcriptional upregulation of *TERT* and adjacent genes distal of the breakpoints, and with substantial epigenetic remodeling of this genomic region [113,115,194]. As described in other entities, the telomeres were relatively short in neuroblastomas in which telomerase was activated [84,113]. In another subset of high-risk tumors, inactivating *ATRX* mutations had been identified, which were invariably associated with ALT [113,195,196,197]. In both *ATRX*-inactivated and *ATRX*-wild-type neuroblastomas bearing ALT activation, elongated telomeres were detected, which was associated with poor outcome in non-*MYCN*-amplified neuroblastoma [175,188,195,198]. Together, all of these studies suggested that neuroblastomas harboring any kind of telomere maintenance exhibit aggressive growth characteristics. In fact, two recent studies demonstrated that telomere maintenance, either conferred by telomerase or ALT activation, is a defining hallmark of high-risk disease in neuroblastoma [113,174]. By contrast, low-risk tumors invariably lacked telomere maintenance mechanisms, presumably leading to the inability of these tumors to gain immortal proliferation capacity, which in turn results in spontaneous regression. Furthermore, patient outcome was particularly poor when tumors harbored mutations affecting the RAS or the p53 pathway in addition to telomere maintenance [174]. By contrast, these mutations had no effect on patient outcome and were compatible with spontaneous regression in the absence of telomere maintenance mechanisms. Together, these studies demonstrated that activation of telomere maintenance mechanisms essentially determines the biological behavior of neuroblastoma, thereby providing a precise mechanistic classification of clinical phenotypes in this malignancy [174].

## 9. Other Embryonal Tumors

### 9.1. Wilms Tumor (Nephroblastoma)

Wilms tumor, also known as nephroblastoma, occurs in young children and is responsible for 95% of malignant kidney tumors in patients younger than 15 years [200]. The Wilms tumor gene (*WT1*) is deleted or mutated to an inactive form in a fraction of Wilms tumors and is a known repressor of *TERT* expression [201]. In 2011, Venturini and colleagues reported on the presence of telomerase activity and ALT in Wilms tumor. They investigated 34 samples from 30 patients, and ALT was detected as the sole mechanism of telomere maintenance in five samples and in association with telomerase activity in six samples. Seventeen samples exhibited telomerase activity only, and in six cases neither of the two mechanisms was found. However, the authors pointed out that the finding of simultaneous activation of telomerase and ALT, although observed in many other neoplasms such as osteosarcoma, liposarcoma, glioblastoma multiforme and diffuse malignant peritoneal mesothelioma, warrants further investigation [184].

### 9.2. Hepatoblastoma

Hepatoblastoma is the most common malignant liver tumor in childhood [202]. Data on telomere maintenance in childhood hepatoblastoma is sparse, and only little progress has been made in the understanding of telomere maintenance mechanisms in hepatoblastoma since Hiyama et al. evaluated telomerase expression and activity in 63 patients in 2004 [179]. In this study, *TERT* mRNA levels were an independent prognostic factor in a multivariate model including histology, stage, gender, age, and completeness of resection. Over 90% of primary tumors showed detectable *TERT* mRNA expression, and telomerase activity was found in 31% (12/39 samples) [179]. Although *TERT* promoter mutations have been described as an early event in hepatocellular carcinoma on cirrhosis, they were not observed in an exome sequencing analysis of 18 classical hepatoblastoma specimens, but appeared to be an exclusive feature of transitional liver cell tumors with clinical and histo-pathological features reminiscent of hepatocellular carcinoma, thereby providing an excellent marker for the detection of high-risk patients [171,203,204]. Furthermore, in a large-scale sequencing study analyzing genomic alterations in childhood cancers in general, no *ATRX* mutations were found in 16 hepatoblastoma samples [109]. However, these analyses do not provide sufficient information to predict the absence of ALT in hepatoblastoma, and further analyses are necessary to fully characterize the whole spectrum of telomere maintenance in this pediatric liver tumor.

### 9.3. Retinoblastoma

Similar to hepatoblastoma, studies on telomerase activity are rare for retinoblastoma. Retinoblastoma is the most common primary intraocular malignancy of childhood [205,206]. Still, the study published by Gupta and colleagues in 1996 is one of the most accurate analyses of telomerase activity in pediatric retinoblastoma, describing active telomerase in 50% of all cases. The telomerase-negative retinoblastomas showed no evidence of ALT [178]. In line with this, novel analyses based on telomeric DNA content in whole genome data found no evidence of ALT in 4 primary pediatric retinoblastomas [62].

## 10. Leukemia

Acute lymphoblastic leukemia (ALL) and acute myeloblastic leukemia (AML) are the two most frequent leukemia types diagnosed in childhood, accounting for about 30% of all malignancies in children younger than 15 years [207,208]. Telomeric DNA content analysis using whole genome sequencing revealed that pediatric leukemia cells have short telomeres with no evidence for ALT, suggesting telomerase-mediated telomere maintenance [62]. In line with this hypothesis, TERT-based T cell-mediated immunotherapy for leukemia treatment has been shown to be a promising strategy in vivo [208].

### 10.1. ALL

ALL occurs approximately five times more frequently than AML and accounts for approximately 25% of all cancers in children [209]. It has long been known that the level of telomerase activity in bone marrow specimens from pediatric ALL patients at diagnosis is significantly higher (10- to 20-fold) compared to levels found in patients in remission or healthy subjects [210]. More recently, analysis of a larger cohort confirmed these findings, showing no evidence for ALT, but increased telomerase activity in 54% of all samples, which was significantly associated with hypermethylation of the *CDKN2B* gene locus [62,186].

### 10.2. AML

Pediatric AML accounts for approximately 20% of childhood leukemias [211]. This heterogeneous malignant disease is characterized by abnormal proliferation and impaired differentiation of myeloid precursor cells [212]. In contrast to adult AML patients, in whom constitutional loss-of-function mutations in telomerase complex genes have been implicated as risk factors, variants in the telomerase complex genes *TERT* and *TERC* are rare and do not seem to be a risk factor for developing pediatric AML [170,213]. Nevertheless, telomerase activity in bone marrow is a highly significant factor for poor prognosis in pediatric AML patients and detectable in about half of all cases [185]. Furthermore, as described for childhood ALL, AML cells were reported to have loss of telomeric DNA, indicating the absence of ALT [62].

## 11. Sarcomas

Sarcomas are a heterogeneous group of tumors and account for about 15% of all cancers in children, adolescents and young adults [214]. In contrast to pediatric leukemia, ALT is relatively common in many subtypes of sarcomas [60,215,216,217]. ALT and telomerase occur in a non-random and tumor-type specific manner that varies significantly between different tumor types.

### 11.1. Ewing Sarcoma

Ewing sarcoma is the second most common malignant bone tumor occurring in children and young adults [218]. This tumor is derived from primordial bone marrow–derived mesenchymal stem cells, and the genetic hallmark of Ewing sarcoma is the recurrent t(11;22)(q24;q12) translocation resulting in the *EWS/FLI1* fusion gene, identified in approximately 90% of all cases [218,219]. Many studies have reported telomerase activity in a substantial subgroup of Ewing sarcomas (about 60%) and suggested to use this significant prognostic variable as a molecular marker for malignancy, whereas ALT seems to be absent in pediatric Ewing sarcomas [60,159,180,220,221,222].

### 11.2. Rhabdomyosarcoma

Distinct telomere maintenance mechanisms have been found in alveolar (ARMS) and embryonal rhabdomyosarcoma (ERMS) [181]. Rhabdomyosarcoma is the most common soft-tissue sarcoma in children and accounts for roughly 4.5% of all childhood cancers [214,223]. In a study from 2008, Ohali and colleagues analyzed telomere maintenance in 31 primary untreated rhabdomyosarcomas, 15 of which were of the alveolar subtype and 16 of the embryonal subtype. The majority of alveolar rhabdomyosarcomas harbored telomerase activity (71%), while ALT was absent in this subtype. By contrast, some embryonal tumors exhibited an ALT or “ALT-like” phenotype lacking telomerase activity (38%), whereas others harbored activated telomerase (50%). In both subtypes, no telomere maintenance mechanism was detected in some tumors. The authors concluded that their findings suggest that ARMS predominantly use telomerase-dependent telomere maintenance, whereas in embryonal tumors both telomerase and ALT may occur [181].

### 11.3. Osteosarcoma

In pediatric osteosarcoma, ALT appears to be the predominant mechanism for telomere maintenance (about 80%), with loss of *ATRX* expression being observed in 30% of all cases [60,224,225]. Osteosarcoma is one of the most common types of bone cancer in children and accounts for nearly 3% of all childhood cancers [226]. Absence of telomere maintenance has been described to be a favorable prognostic factor [224]. However, in approximately 30% of pediatric cases, telomerase activity has also been found, with a high number of tumors showing both telomerase and ALT activity [182]. Although it has been proposed that telomerase and ALT activity can coexist in certain human cells, further studies are necessary to validate these findings and determine their relevance for osteosarcoma pathogenesis [227].

## 12. Brain Tumors

Currently, brain tumors are the major cause of mortality and long-term morbidity in pediatric oncology [228]. In recent years, many studies have investigated various aspects of telomere maintenance in this very heterogeneous group of tumors that have extremely divergent clinical courses and outcomes [183,229,230,231,232].

### 12.1. Medulloblastoma

Medulloblastoma is the second most common pediatric brain tumor, accounting for approximately 20% of all primary central nervous system tumors in children between the ages of 0 and 14 years and 6% of those occurring in patients aged 15–19 years [233,234]. *MYC*-driven medulloblastomas have a particularly poor prognosis, as they are commonly metastatic and resistant to standard therapy [235]. With more than 80% of all samples positive for telomerase activity, this type of tumor has one of the highest rates of telomerase-dependent telomere maintenance in childhood cancers [183]. ALT is detected in a small subgroup, but appears to play a minor role in primary medulloblastoma [180]. In contrast to this observation, a recent analysis of 43 pediatric metastatic medulloblastomas revealed that these cases frequently activate the ALT pathway, suggesting that it ALT may be a common process to escape senescence in primary medulloblastomas that metastasize [236].

### 12.2. Glioma

Gliomas vary from low grade tumors (WHO Grade I) to high grade glioblastomas (WHO Grade IV) [237]. Pediatric low-grade gliomas (PLGGs), classified as WHO Grades I and II, are the most common brain tumors observed in childhood and represent 30–50% of all central nervous system neoplasms among children [238]. Clinically, these tumors are characterized by slow growth rates, a low risk of malignant transformation to high-grade glioma, and relatively favorable long-term survival rates [239,240,241]. Telomerase activity has been reported to be absent in PLGGs [183,242]. Furthermore, little evidence was found for ALT in these tumors, with only one sample being positive in a study comprising 60 tumor specimens [180]. In contrast, about half of all pediatric high grade gliomas (PHGGs) is positive for ALT, and a smaller subgroup of about 23% has activated telomerase, with *TERT* promoter mutations and chromothripsis being quite uncommon [104,180,199,243]. Adult high-grade gliomas show the reverse pattern [243].

### 12.3. Ependymoma

Ependymomas are the third most common type of brain tumors in children (following astrocytoma and medulloblastoma), accounting for 6–12% of brain tumors in children [244,245]. Spinal ependymoma is linked with an indolent clinical course and good prognosis, whereas intracranial ependymoma is associated with an aggressive clinical course and poor prognosis [246]. In a comprehensive study using three different methods to determine ALT status (terminal restriction fragment analysis, telomere fluorescence in situ hybridization and C-circle assay), ALT was absent in all 95 tumor specimens investigated [180]. In contrast, telomerase activity was detected in the majority (64%) of cases [176].

In other major childhood cancers, such as lymphoma or germ cell tumors, telomere maintenance mechanisms have only been investigated in studies that are too small or exploratory to allow for general conclusions on their prevalence [68,104,247,248,249,250,251,252,253].

Taken together, assessment of telomere maintenance mechanisms may contribute to the implementation of biomarker-driven risk estimation and treatment stratification not only in neuroblastoma or pediatric cancers, but in the field of oncology in general. In addition, development of more specific and effective inhibitors targeting telomerase or the ALT pathway may provide options for novel therapeutic strategies in difficult-to-treat pediatric tumors [254].

## 13. Future Perspectives

While substantial progress has been made in determining the prevalence and clinical relevance of telomerase activation and ALT in childhood cancers over recent years, it is evident that our knowledge of telomere maintenance in pediatric oncology is still limited. The data shown in Table 1 suggest that considerable fractions of highly aggressive malignancies in childhood may lack telomere maintenance mechanisms (e.g., ALL or Ewing sarcoma). While this possibility cannot formally be excluded, it appears more likely, however, that the reported low frequencies are due to technological limitations in detecting telomere maintenance. On the other hand, cases bearing activated telomerase and ALT sum up to >100% in other cancer types (Wilms tumor and osteosarcoma), suggesting co-existence of these mechanisms. It remains to be determined, though, whether this finding is related to methodological issues, or whether indeed telomerase and ALT co-exist in some malignant entities.

The prototypic example of neuroblastoma demonstrates that in-depth knowledge of telomere maintenance mechanisms may provide insights into the biological behavior of cancer, and may be useful for precise risk estimation and treatment stratification in the future. In addition, the fact that telomerase and ALT are active in the vast majority of human cancers, while being absent in most normal tissues, offers starting points for therapeutic strategies targeting telomere maintenance mechanisms. In fact, several treatment approaches, such as nucleosides analogs, oligonucleotides, small molecule inhibitors, G-quadruplex stabilizers, immunotherapy, gene therapy, molecules that affect the telomere/telomerase associated proteins, agents from microbial sources and many others, have been proposed to inhibit cancer cell growth by targeting telomerase (reviewed in [255]). Major limitations of previous therapeutic attempts have been insufficient information on the ultra-structural resolution of the telomerase holoenzyme and lack of knowledge of the precise mechanism of ALT. Both of these issues, however, are in the focus of current research activities and may be solved over the next few years, thereby providing a sound basis for the development of effective therapies targeting telomere maintenance mechanisms in human cancer [154,155,256].

## Figures and Tables

**Table 1 ijms-20-05836-t001:** Telomere biology in pediatric cancers.

Cancer Type	Telomerase Positive Cases ^1^ in %	ALT Positive Cases ^2^ in %
ALL	54 (39/72) [186]	0 (0/53) [62]
AML	55 (22/40) [185]	0 (0/17) [62]
Medulloblastoma	86 (6/7) [183]	2 (3/137) [180]
Glioma (high grade)	23 (5/22) [199]	44 (14/32) [159]
Glioma (low grade)	0 (0/11) [183]	2 (1/60) [180]
Ependymoma	64 (23/36) [176]	0 (0/95) [180]
Neuroblastoma	38 (78/208) [174]	15 (31/208) [174]
Wilms tumor	72 (23/32) [184]	35 (11/31) [184]
Hepatoblastoma	31 (12/39) [179]	Not evaluated
Retinoblastoma	50 (17/34) [178]	0 (0/4) [62]
Osteosarcoma	32 (14/44) [182]	80 (35/44) [182]
Ewing Sarcoma	64 (9/14) [60]	0 (0/14) [60]
Rhabdomyosarcoma (ERMS)	50 (8/16) [181]	38 (6/16) [181]
Rhabdomyosarcoma (ARMS)	71 (10/14) [181]	0 (0/15) [181]

^1^ High *TERT* RNA expression and/or active enzyme measured by Telomere Repeat Amplification Protocol (TRAP). ^2^ High telomeric DNA content, C-circle analysis or telomere restriction fragment (TRF) analysis.

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
