# Peer review of "Telomere Maintenance in Pediatric Cancer"

_ijms, 2019, doi:10.3390/ijms20235836_

Round 1

Reviewer 1 Report

“Telomere maintenance in pediatric cancers” is a well written manuscript; however, it can be improved with some modifications.

One of the shortcomings in the review pertains to the lack of details about pediatric cancers in general and distinguishing features compared to adult cancers. The placement of telomerase within that context will improve the quality. Over half of the review is spent on describing telomerase in general and then the telomere maintenance in pediatric cancers starts abruptly.

The references associated with a number of critical statements in the manuscript are actually of reviews. It is prudent that instead of reviews, the original papers describing those results are cited. For example on page 2, it is stated that ‘Abnormally long telomeres in somatic cells have been linked to numerous diseases such as familial melanoma, familial chronic lymphocytic leukemia’. The two references given in support are reviews. Authors should refer to original reports, which have consistently shown association between long telomeres and increased risk of several cancers not just familial as shown in reference 7. The problem with references is throughout as authors have relied mostly on reviews instead of original papers.

While mentioning the TERT promoter mutations first time, the two original discovery papers (currently references 134 and 135) should be cited as those were the first studies to describe those mutations.

The nomenclature of the TERT promoter mutations should be changed from C228T and C250T to -124 C>T and -146 C>T. although the nomenclature used by authors is used in literature but it is scientifically incorrect. The 228 and 250 numbers refer to the last digits of genomic coordinates from the human genome version, which is no longer used. -124 C>T and -146 C>T captures the positions from the ATG start site of the TERT gene.

Author Response

Point-to-point response
Reviewer: 1
“Telomere maintenance in pediatric cancers” is a well written manuscript; however, it can be improved with some modifications.
One of the shortcomings in the review pertains to the lack of details about pediatric cancers in general and distinguishing features compared to adult cancers. The placement of telomerase within that context will improve the quality. Over half of the review is spent on describing telomerase in general and then the telomere maintenance in pediatric cancers starts abruptly.

Response: We thank the reviewer for this comment. To address this issue, we have substantially shortened the text passages describing telomerase in general, added information about telomere maintenance data from pediatric studies and point to the differences between childhood and adult cancers.

The references associated with a number of critical statements in the manuscript are actually of reviews. It is prudent that instead of reviews, the original papers describing those results are cited. For example on page 2, it is stated that ‘Abnormally long telomeres in somatic cells have been linked to numerous diseases such as familial melanoma, familial chronic lymphocytic leukemia’. The two references given in support are reviews. Authors should refer to original reports, which have consistently shown association between long telomeres and increased risk of several cancers not just familial as shown in reference 7. The problem with references is throughout as authors have relied mostly on reviews instead of original papers.

Response: We thank the reviewer for suggesting to present more original data instead of reviews. We therefore replaced the vast majority of review references with original reports and provided detailed information on original results, not only in the context of association between long telomeres and increased risk of several cancers, but throughout the complete manuscript. Very few exceptions were kept in the text and marked (“as reviewed in”).

While mentioning the TERT promoter mutations first time, the two original discovery papers (currently references 134 and 135) should be cited as those were the first studies to describe those mutations.

Response: According to the reviewer’s suggestion, the two original discovery papers were cited in the text in the passage where the TERT promoter mutations were mentioned for the first time.
The nomenclature of the TERT promoter mutations should be changed from C228T and C250T to -124 C>T and -146 C>T. although the nomenclature used by authors is used in literature but it is scientifically incorrect. The 228 and 250 numbers refer to the last digits of genomic coordinates from the human genome version, which is no longer used. -124 C>T and -146 C>T captures the positions from the ATG start site of the TERT gene.

Response: According to the reviewer’s suggestion, the nomenclature of the TERT promoter mutations was changed from C228T and C250T to -124 C>T and -146 C>T.

Reviewer 2 Report

The present review summarizes the recent development in the field of telomeres, aging, and cancer. It highlights the key advances that have improved our knowledge of telomeres in pediatric cancer particularly. I am aware of the primary aim of this review is to focus on the pediatric cancer rather than cancer or aging per se. It is however always beneficial to have a short summary of the comparisons of the roles of telomeres in aging, cancer, and pediatric cancer. Are there any potential differences in the clinical practices should telomeres testing were to be implemented in clinics.

Author Response

Reviewer: 2
The present review summarizes the recent development in the field of telomeres, aging, and cancer. It highlights the key advances that have improved our knowledge of telomeres in pediatric cancer particularly. I am aware of the primary aim of this review is to focus on the pediatric cancer rather than cancer or aging per se. It is however always beneficial to have a short summary of the comparisons of the roles of telomeres in aging, cancer, and pediatric cancer. Are there any potential differences in the clinical practices should telomeres testing were to be implemented in clinics.

Response: We thank the reviewer for this comment. We have modified the manuscript accordingly and added summarizing text passages to the sections describing aging (association of telomere length with aging and all-cause mortality), cancer in general (dependence of cancer cells on telomere maintenance for immortalization and malignant behavior) and pediatric cancer in particular (Table 1 is summarizing telomere maintenance data in pediatric cancer types, and more detailed information about the definition of “telomerase/ALT positive tumors" was added as footnotes).

Furthermore, to address the question about expected differences in clinical practices, we have added the following passage to the manuscript: “Taken together, assessment of telomere maintenance mechanisms may contribute to the implementation of biomarker-driven risk estimation and treatment stratification not only in neuroblastoma or pediatric cancers, but in the field of oncology in general. In addition, development of more specific and effective inhibitors targeting telomerase or the ALT pathway may provide options for novel therapeutic strategies in difficult-to-treat pediatric tumors.“

Reviewer 3 Report

Ackermann and Fischer perform a review on the role of telomere maintenance in childhood cancers. Although their citation list is lengthy, too much of the paper is spent reviewing general features of telomere biology independent of cancer. They next discuss telomeres and cancer but rely too heavily on adult cancer literature, which is certainly richer than the pediatric literature, but there's plenty of pediatric literature they could use/cite instead. Specific instances of incorrect statements or references are noted below, as are some places where expanded text is necessary. 

1) "Telomerase activity is well correlated with TERT expression, and ectopic expression of TERT expression in telomerase negative cells is sufficient to confer telomerase activity, suggesting that TERT is the rate-limiting component of telomerase activity in most cells". This is true, with one major exception that seems especially pertinent to your topic: TERC appears to be the rate-limiting factor for telomerase activity in the developing embryo during gestation.

2)  When discussing the C228T and C250T mutations, the authors state "These genetic alterations create DNA binding sites for members of the E26 transformation-specific transcription factor family, which promote an increase TERT expression [134-136]." The TF is actually GABPB1L. These citations identified the mutations but did little to functionally demonstrate that an ETS factor bound here and activated telomerase. This Science paper demonstrated the mutations recruited an ETS TF - PMID: 25977370. This Cancer Cell paper showed the specific isoform (PMID: 30205050)

3) "To date, there has been no appropriate clinical study providing evidence that telomere length can reliably predict function decline in aging individuals or may actually predict morbidity or mortality". Such a study would not be clinical, it would be epidemiological. This is the closest thing I am aware of PMID: 28459963. 

4) "Abnormally long telomeres in somatic cells have been linked to numerous diseases such as familial melanoma, familial chronic lymphocytic leukemia, a Li-Fraumeni-like syndrome [27, 49] and an increased cancer risk in general [38, 50, 51]." This whole sentence seems to be about germline POT1/shelterin mutations, which cause familial melanoma, glioma, and LFLS with angiosarcomas. The citations are of reviews that don't really cover these topics in-depth anyways. I'd suggest replacing the whole sentence with something like this: "Longer telomere length appears to increase risk of numerous cancer types [PMID: 28241208], especially melanoma [PMID: 25231748], adult glioma [PMID: 26646793], and chronic lymphocytic leukemia [PMIDs: 27197291 and 27008888]. Similar analyses in childhood cancers have observed association between longer telomere length and risk of neuroblastoma [PMID: 27207662] and osteosarcoma [PMID: 31525475]."

5) "High mutation frequencies were found in nine cancer types, including melanoma, liposarcoma, hepatocellular carcinoma, urothelial carcinoma, squamous cell carcinoma of the tongue, medulloblastoma, and glioma subtypes (including 83% of primary glioblastoma, the most common brain tumor type) [137]." Specify you're talking about "adult glioma". 

6) "In general, TERT promoter mutations seem to mainly represent characteristic alterations of adult cancers and to occur infrequently in pediatric cancers and tumors of young adults [129]". - add - "with the exception being SHH-activated medulloblastoma, especially those tumors diagnosed in adolescents and young adults."

7) "One hypothesis (and non-mutually exclusive mechanism) is that promoter hypermethylation can open the promoter proximal chromatin conformation and result in increased transcription of the locus [140,
210 161]"... -add- "...due to the high GC content of the TERT promoter and its propensity to adopt G-quadruplex conformations."

8) "For example, in hepatoblastoma, the most common childhood liver cancer, as well as in pediatric gliomas, acute myeloid leukemia, thyroid cancers and medulloblastoma, TERT gene or promoter mutations are not observed or far less frequent than in corresponding adult malignancies [132, 141]." This is not true of medulloblastoma. First, there really isn't adult medulloblastoma short of a few case-reports. Second, medulloblastoma has high rates of TERT promoter mutations.

9) Table 1 is based on such small numbers in many rows that these really should be presented as point estimates with confidence intervals using a forest plot format. It's also a problem that 'telomerase postive" really isn't defined anywhere and is a very diverse assessment across samples. For instance, a tumor with a C250T mutation is probably "telomerase positive", as is a tumor with high TERT RNA expression, as is a tumor where TERT protein levels were measured. This varies from paper to paper. 

10) Lymphomas and germ cell tumors are major childhood cancers that have not been included in your review. 

11) "About 18% to 20% of childhood brain tumors are medulloblastomas, with the MYC-subgroup being one of the most aggressive pediatric brain tumors". There is no MYC-subgroup. The subgroups of MB are: WNT, SHH, Group 3, Group 4. 

12) ALT is detected in a small subgroup, but appears to play a minor role in primary medulloblastoma [228]. In contrast to this observation, a recent analysis of 43 pediatric metastatic medulloblastomas revealed that these cases frequently activate the ALT pathway, suggesting that it ALT may be a common process to escape senescence in primary metastatic medulloblastomas [270]." Although included in the title of the paper you cite, "primary metastatic medulloblastoma" is an uncommon terminology without additional context. These should be referred to as "primary medulloblastomas that metastasize". 

13) Spontaneous tumor regression is a unique feature of pediatric low grade gliomas, in which telomerase activity has been reported to be absent". This is an incredibly rare phenomenon with fewer than 20 case-reports outside the NF1 syndrome context. In fact, this is primarily a feature of low-risk neuroblastoma, which would be a much more sensible place to discuss this. 

14) "In contrast, about half of all pediatric high grade gliomas have activated telomerase and a smaller, but significant subgroup of about 20% is positive for ALT [228, 229]." This is flipped. ALT is common and TERT reactivation is rare in pediatric HGG, with TERT promoter mutations and chromothripsis being quite uncommon. If your references here also say that telomerease-based telomere maintenance is more common that ALT in pediatric high grade glioma, then the papers are anomalies and should be replaced with more current or larger studies. 

15) A section on the association of TERT activation (especially TERT promoter mutations) and age-at-diagnosis in pediatric cancers would be useful.

16) Almost no talk of telomere maintenance and pediatric patient prognosis is presented. If this is intentional, then it probably should be stated upfront that you're discussing only tumor initiation. 

Author Response

Point-to-point response
Reviewer: 3

Ackermann and Fischer perform a review on the role of telomere maintenance in childhood cancers. Although their citation list is lengthy, too much of the paper is spent reviewing general features of telomere biology independent of cancer. They next discuss telomeres and cancer but rely too heavily on adult cancer literature, which is certainly richer than the pediatric literature, but there's plenty of pediatric literature they could use/cite instead. Specific instances of incorrect statements or references are noted below, as are some places where expanded text is necessary.

Response: We thank the reviewer for this comment. To address this issue, we have substantially revised the manuscript, shortened text passages describing telomere biology independent of cancer and in adult cancers, added information about telomere maintenance data from pediatric studies and highlighted differences between childhood and adult cancers.

1) "Telomerase activity is well correlated with TERT expression, and ectopic expression of TERT expression in telomerase negative cells is sufficient to confer telomerase activity, suggesting that TERT is the rate-limiting component of telomerase activity in most cells". This is true, with one major exception that seems especially pertinent to your topic: TERC appears to be the rate-limiting factor for telomerase activity in the developing embryo during gestation.

Response: We agree with the reviewer that TERC appears to be the rate-limiting factor for telomerase activity in the developing embryo during gestation and have modified the text accordingly (“Telomerase activity is well correlated with TERT expression, and ectopic expression of TERT expression in telomerase negative cells is sufficient to confer telomerase activity, suggesting that TERT is the rate-limiting component of telomerase activity in most cells with few exceptions, such as the developing embryo during gestation.”).

2) When discussing the C228T and C250T mutations, the authors state "These genetic alterations create DNA binding sites for members of the E26 transformation-specific transcription factor family, which promote an increase TERT expression [134-136]." The TF is actually GABPB1L. These citations identified the mutations but did little to functionally demonstrate that an ETS factor bound here and activated telomerase. This Science paper demonstrated the mutations recruited an ETS TF - PMID: 25977370. This Cancer Cell paper showed the specific isoform (PMID: 30205050)

Response: We thank the reviewer for suggesting to present more specific data and cite more specific references. We have modified the manuscript accordingly and added the suggested information and publications (“These genetic alterations create DNA binding sites that specifically recruit the multimeric E26 transformation-specific factor GABP, leading to increased TERT expression [PMIDs: 25977370, 30205050 and 25261935].”).

3) "To date, there has been no appropriate clinical study providing evidence that telomere length can reliably predict function decline in aging individuals or may actually predict morbidity or mortality". Such a study would not be clinical, it would be epidemiological. This is the closest thing I am aware of PMID: 28459963.

Response: We agree with the reviewer that such a study would be epidemiological and have modified the text accordingly. The word “clinical” is no longer used in the new version of the sentence and the suggested publication was added (“While leukocyte telomere length has been shown to be significantly associated with all-cause mortality, no study has provided evidence that telomere length can reliably predict function decline in old people, or may actually predict age, morbidity or mortality in individuals [PMIDs: 28459963 and 30094266].”).

4) "Abnormally long telomeres in somatic cells have been linked to numerous diseases such as familial melanoma, familial chronic lymphocytic leukemia, a Li-Fraumeni-like syndrome [27, 49] and an increased cancer risk in general [38, 50, 51]." This whole sentence seems to be about germline POT1/shelterin mutations, which cause familial melanoma, glioma, and LFLS with angiosarcomas. The citations are of reviews that don't really cover these topics in-depth anyways. I'd suggest replacing the whole sentence with something like this: "Longer telomere length appears to increase risk of numerous cancer types [PMID: 28241208], especially melanoma [PMID: 25231748], adult glioma [PMID: 26646793], and chronic lymphocytic leukemia [PMIDs: 27197291 and 27008888]. Similar analyses in childhood cancers have observed association between longer telomere length and risk of neuroblastoma [PMID: 27207662] and osteosarcoma [PMID: 31525475]."

Response: We thank the reviewer for optimizing this text passage and have modified the manuscript as suggested (“Longer telomeres appear to increase the risk of numerous cancer types PMID: 28241208], especially melanoma [PMID: 25231748], adult glioma [PMID: 26646793], and chronic lymphocytic leukemia [PMIDs: 27197291 and 27008888]. Similar analyses in childhood cancers have observed associations
between longer telomeres and the risk of neuroblastoma [PMID: 27207662] and osteosarcoma [PMID: 31525475].”).

5) "High mutation frequencies were found in nine cancer types, including melanoma, liposarcoma, hepatocellular carcinoma, urothelial carcinoma, squamous cell carcinoma of the tongue, medulloblastoma, and glioma subtypes (including 83% of primary glioblastoma, the most common brain tumor type) [137]." Specify you're talking about "adult glioma".

Response: We completely agree with the reviewer’s comment that it is necessary to specify in this sentence and have added the word “adult” as suggested (“High mutation frequencies were found in nine cancer types, including melanoma, liposarcoma, hepatocellular carcinoma, urothelial carcinoma, squamous cell carcinoma of the tongue, medulloblastoma, and adult glioma subtypes (including 83% of primary glioblastoma, the most common brain tumor type).”).

6) "In general, TERT promoter mutations seem to mainly represent characteristic alterations of adult cancers and to occur infrequently in pediatric cancers and tumors of young adults [129]". – add – "with the exception being SHH-activated medulloblastoma, especially those tumors diagnosed in adolescents and young adults."

Response: We thank the reviewer for suggesting this valuable complement and have added this information in the new version of the sentence (“In pediatric cancers and tumors of young adults, these alterations occur at a much lower overall-frequency of about 2.5%, with the exception of sonic-hedgehog-activated (SHH) medulloblastoma, especially those tumors diagnosed in adolescents and young adults.”).

7) "One hypothesis (and non-mutually exclusive mechanism) is that promoter hypermethylation can open the promoter proximal chromatin conformation and result in increased transcription of the locus [140, 210 161]"... -add- "...due to the high GC content of the TERT promoter and its propensity to adopt G-quadruplex conformations."

Response: We thank the reviewer for this comment and have added this information as suggested in a new version of the sentence (“One hypothesis (and non-mutually exclusive mechanism) is that promoter hypermethylation can open the promoter proximal chromatin conformation, due to the high GC content of the TERT promoter and its propensity to adopt G-quadruplex conformations, and thus result in increased transcription of the locus.”).

8) "For example, in hepatoblastoma, the most common childhood liver cancer, as well as in pediatric gliomas, acute myeloid leukemia, thyroid cancers and medulloblastoma, TERT gene or promoter mutations are not observed or far less frequent than in corresponding adult malignancies [132, 141]." This is not true of medulloblastoma. First, there really isn't adult medulloblastoma short of a few case-reports. Second, medulloblastoma has high rates of TERT promoter mutations.

Response: We agree with the reviewer and have provided a new version of the sentence without mentioning medulloblastoma (“For example, in hepatoblastoma, the most common childhood liver cancer, as well as in pediatric gliomas, acute myeloid leukemia and thyroid cancers, TERT gene or promoter mutations have not been observed or far less frequently than in corresponding adult malignancies.”).

9) Table 1 is based on such small numbers in many rows that these really should be presented as point estimates with confidence intervals using a forest plot format. It's also a problem that 'telomerase postive" really isn't defined anywhere and is a very diverse assessment across samples. For instance, a tumor with a C250T mutation is probably "telomerase positive", as is a tumor with high TERT RNA expression, as is a tumor where TERT protein levels were measured. This varies from paper to paper.

Response: We thank the reviewer for this comment and agree that the forest plot format is a good visualization method for meta-analyses. However, because of the lack of standardization in experimental procedures and limited comparability of many data sets, we prefer to cite only the most relevant study. Thus, representative values without confidence intervals are shown. This information was added to the text (“Table 1 shows representative results obtained in studies focusing on initial tumor samples.”). “Telomerase positive” is defined as high TERT RNA expression and/or active enzyme measured by Telomere Repeat Amplification Protocol (TRAP), and “ALT positive” is defined as high telomeric DNA content, C-circle analysis or telomere restriction fragment (TRF) analysis. This information was added as footnotes to Table 1.
10) Lymphomas and germ cell tumors are major childhood cancers that have not been included in your review.

Response: We completely agree with the reviewer’s comment that lymphomas and germ cell tumors are major childhood cancers and should be mentioned in this review. However, data is too sparse (and/or subgroup-specific) and no conclusions can be drawn to date. We have added this information to the manuscript (“In other major childhood cancers, such as lymphoma or germ cell tumors, telomere maintenance mechanisms have only been investigated in studies that are too small or exploratory to allow for general conclusions on their prevalence.”).

11) "About 18% to 20% of childhood brain tumors are medulloblastomas, with the MYC-subgroup being one of the most aggressive pediatric brain tumors". There is no MYC-subgroup. The subgroups of MB are: WNT, SHH, Group 3, Group 4.

Response: We agree with the reviewer’s comment and have re-written this text passage without term “subgroup” (“MYC-driven medulloblastomas have a particularly poor prognosis, as they are commonly metastatic and resistant to standard therapy.”).

12) ALT is detected in a small subgroup, but appears to play a minor role in primary medulloblastoma [228]. In contrast to this observation, a recent analysis of 43 pediatric metastatic medulloblastomas revealed that these cases frequently activate the ALT pathway, suggesting that it ALT may be a common process to escape senescence in primary metastatic medulloblastomas [270]." Although included in the title of the paper you cite, "primary metastatic medulloblastoma" is an uncommon terminology without additional context. These should be referred to as "primary medulloblastomas that metastasize".

Response: We thank the reviewer for this comment and have changed the wording according to the suggestion (“In contrast to this observation, a recent analysis of 43 pediatric metastatic medulloblastomas revealed that these cases frequently activate the ALT pathway, suggesting that it ALT may be a common process to escape senescence in primary medulloblastomas that metastasize.”).

13) Spontaneous tumor regression is a unique feature of pediatric low grade gliomas, in which telomerase activity has been reported to be absent". This is an incredibly rare phenomenon with fewer than 20 case-reports outside the NF1 syndrome context. In fact, this is primarily a feature of low-risk neuroblastoma, which would be a much more sensible place to discuss this.

Response: We completely agree with the reviewer that spontaneous tumor regression is primarily a feature of low-risk neuroblastoma and focus on this in the neuroblastoma section. The text about glioma has been re-written according to the suggestion, without mentioning the rare phenomenon of spontaneous tumor regression in this tumor type (“Pediatric low‐grade gliomas (PLGGs), classified as WHO Grades I and II, are the most common brain tumors observed in childhood and represent 30–50% of all central nervous system neoplasms among children. Clinically, these tumors are characterized by slow growth rates, a low risk of malignant transformation to high-grade glioma, and relatively favorable long‐term survival rates.”).

14) "In contrast, about half of all pediatric high grade gliomas have activated telomerase and a smaller, but significant subgroup of about 20% is positive for ALT [228, 229]." This is flipped. ALT is common and TERT reactivation is rare in pediatric HGG, with TERT promoter mutations and chromothripsis being quite uncommon. If your references here also say that telomerease-based telomere maintenance is more common that ALT in pediatric high grade glioma, then the papers are anomalies and should be replaced with more current or larger studies.

Response: We thank the reviewer for pointing to these non-representative references and data sets. We have replace these studies with more appropriate data sets from larger and more current analyses, hoping to give a more reliable overview of telomere maintenance data in pediatric HGG in the revised manuscript. The text and table 1 have been updated accordingly (“In contrast, about half of all pediatric high grade gliomas (PHGGs) is positive for ALT, and a smaller subgroup of about 23% has activated telomerase, with TERT promoter mutations and chromothripsis being quite uncommon.”).

15) A section on the association of TERT activation (especially TERT promoter mutations) and age-at-diagnosis in pediatric cancers would be useful.

Response: We totally agree with the reviewer that analyzing the association of TERT or telomere maintenance activation and age-at-diagnosis in pediatric cancers in general would be extremely interesting and useful. However, currently available data is not sufficient and often not detailed enough to draw conclusions due to methodological and technical limitations (different experimental protocols and platforms, non-standardized data normalization methods, batch- or analysis-specific thresholds and many more). In term of TERT promoter mutations alone, however, some large data set from pediatric sequencing cohorts have been published, and we have added this information in the section about TERT promoter mutations (“In general, TERT promoter mutations appear to mainly represent characteristic alterations of adult cancers, and their occurrence is strongly correlated with age. In pediatric cancers and tumors of young adults, these alterations occur at a much lower overall-frequency of about 2.5%, with the exception of sonic-hedgehog-activated (SHH) medulloblastoma, especially those tumors diagnosed in adolescents and young adults. Data on age distribution of TERT promoter mutations in pediatric cancers is rare, but a few studies suggest a lower frequency in infants as compared to older children [PMIDs: 25231549 and 24154961].”).

16) Almost no talk of telomere maintenance and pediatric patient prognosis is presented. If this is intentional, then it probably should be stated upfront that you're discussing only tumor initiation.

Response: We absolutely agree with the reviewer that the association of telomere maintenance and prognosis in most pediatric cancers is poorly understood. In neuroblastoma (PMIDs: 26466568, 30523111, 21319260, 16917952, 26653081, 27793328, 22416102 and 25487495) and a few other entities (PMIDs: 15280920, 15365076, 12712473, 25229770 and 14551302) there is a clear correlation of any kind of telomere maintenance with poor prognosis, which we are presenting in the revised manuscript. However, for the majority of pediatric cancers, no or little information is available to date. Given the strong impact of telomere maintenance on clinical courses in some cancers, such as neuroblastoma, we feel that investigation of telomere maintenance in large and clinically well-annotated cohorts should be a major task of future studies.

Round 2

Reviewer 1 Report

No comments

Reviewer 3 Report

Authors have made adequate changes